# Assessment of Surgical Difficulty in Patients with Rectal Cancer—The Impact of Pelvimetry

João Stuart [1,†], Pedro Miguel Dias dos Santos [2,*,†,‡], Carlos Costa Pereira [3] and Sandra F. Martins [3,4,5,*]

1   Nova Via Family Health Unit, 4405-535 Porto, Portugal; joaostuart05@gmail.com
2   Torres Vedras's Hospital, 2560-295 Torres Vedras, Portugal
3   Coloproctology Unit, Braga's Hospital, 4710-243 Braga, Portugal; carlos.pereira@hb.min-saude.pt
4   Life and Health Science Research Institute (ICVS), School of Medicine, University of Minho, 4710-057 Braga, Portugal
5   Life and Health Sciences Research Institute (ICVS), 3B's-PT Government Associate Laboratory, 4806-909 Guimarães, Portugal
*   Correspondence: pedro.m.santos@choeste.min-saude.pt (P.M.D.d.S.); sandramartins@med.uminho.pt (S.F.M.)
†   These authors contributed equally to this work.
‡   Dr. Pedro Miguel Dias dos Santos is an internship in General Surgery at Braga's Hospital.

**Abstract:** Background: Low-quality tumoral surgical excision is the major relapse factor in rectal cancer. If the surgery is highly difficult, the quality of the resection might be compromised. In the literature, it is described how low pelvic dimensions can make this type of surgery difficult. The main aim was to study the influence of pelvic measures in surgical difficulty on the patients submitted to tumoral surgical resection with curative intent. Methods: A retrospective, observational and analytic study was conducted. A total of 73 patients over a period of 3 years were included. Demographic and surgical data, as well as measurements of the pelvis taken from MRI, were collected. An univariate and multivariate analysis was performed. Results: 11 (15.1%) patients were classified as having highly difficult surgeries. All 11 patients were male. Significant differences were found between groups regarding gender ($p = 0.013$), transverse diameter of the pelvis ($p < 0.001$), interspinal distance ($p = 0.014$) and intertuberous distance ($p < 0.001$). The logistic regression revealed that a small transverse diameter (O.R. 0.919, 95% I.C. 0.846–0.999, $p = 0.047$) increases the degree of difficulty of the surgery. Conclusions: Male patients with a small pelvic measurement deserve a thorough surgical plan that predicts a quality resection.

**Keywords:** magnetic resonance imaging; pelvimetry; rectal cancer; surgical difficulty

## 1. Introduction

Colorectal cancer (CRC) is the third most common cancer worldwide, and rectal cancer (RC) accounts for approximately one third of these cancers [1,2].

Tumor staging is one of the most decisive steps for the prognosis and correct therapeutic planning of patients with CRC, with some particularities in RC. The staging system used is the American Joint Committee on Cancer (AJCC) TNM, which describes the tumor in relation to its size and the invasion of surrounding structures (T), the number of invaded lymph nodes (N), and the presence or absence of metastases (M) [3]. Currently, staging in RC is performed using Magnetic Resonance Imaging (MRI), which allows a more precise characterization of the relationship between the mesorectal fascia and the tumor margin [4,5].

The goal of rectal cancer treatment is to optimize disease-free and overall survival while preserving function and minimizing the risk of local recurrence and toxicity from both radiation and systemic therapy.

Currently, the first-line treatment is anterior resection of the tumor, accompanied by total excision of the mesorectum (TEM), which is the lymphovascular fatty tissue (mesorectum) that surrounds the rectum, between the visceral fascia and the pelvic fascia [6–9].

The decision to treat a patient with neoadjuvant radiochemotherapy is dependent on the clinical tumor stage at presentation.

Because the pelvic cavity is a potentially narrow structure, with little space for the surgical technique and the surgeon's visualization, this type of procedure is more difficult [10,11]. This is particularly true in low rectal tumor resection surgeries [12]. In addition to a narrow pelvis, which is visualized by MRI, a high Body Mass Index (BMI), greater proximity of the tumor to the anal margin (AM), the presence of preoperative treatment and tumor staging are other factors that add difficulty [8,12,13].

Although the assessment of surgical difficulty generally has a subjective component of the surgeon's experience, several factors are described in the literature as indicators of the degree of surgical difficulty. One study brought these factors together and created a scoring system (Table 1) that divides patients into low or high surgical difficulty. A score of 6 or higher in this system classifies the surgery as difficult [11].

**Table 1.** Surgical difficulty criteria.

|  | Score |
| --- | --- |
| Duration of surgery > 300 min | 3 points |
| Conversion to open procedure | 3 points |
| Use transanal dissetion | 2 points |
| Postoperative hospital stay > 15 days | 2 points |
| Blood loss > 200 ml | 1 point |
| Morbidity grade II and III (Clavien–DIndo classification) | 1 point |

Escal et al., when applying this system, developed the following predictive model to identify high-risk patients: high surgical difficulty = $0.99 \times$ mesorectal fat area (MFA) $- 1.15 \times$ coloanal anastomosis (CAA) $- 1.05 \times$ intertuberous distance (ITD) $+ 1.38 \times$ BMI [11]. Each of these variables was assigned 1 point in cases where the MFA is greater than 20.7 cm$^2$, when there was a need for CAA, when the ITD is less than 10.1 cm and when the BMI is greater than 30 kg/m$^2$ [11]. However, there were contradictory results regarding the impact of these factors on surgical difficulty, leading to great discussion among specialists [8,9].

Thus, we know that a low-quality surgical tumor excision is the main cause of recurrence and that if a tumor resection surgery has a high degree of difficulty, its quality may be compromised. It is important to know in advance which patients will present a high level of surgical difficulty so that surgeons can decide the best way to approach these tumor resections. It is therefore extremely interesting to verify if current surgical difficulty prediction models can be applied to our hospital population, enabling the creation of a model that is better suited to our population, if these ones do not prove to be adequate. In this way, it will be possible to establish the prognosis of these patients in a simple and efficient way and offer them a personalized approach.

The main aim is to study the influence of pelvic measures in surgical difficulty on the patients submitted to tumoral surgical resection with curative intent. The secondary aims are to apply the current predictor model to our hospital population with RC, verify its validity and, if possible, design a better-suited predictive model of surgical difficulty.

## 2. Materials and Methods

An observational, analytic, retrospective and descriptive study was conducted. This study used a non-random convenience sample, and several inclusion and exclusion criteria were applied. The target population consisted of the patients submitted to anterior resection for RC, performed in Braga's Hospital, between March 2016 and July 2019.

The inclusion criteria were as follows:

- Patients diagnosed with RC up to 10 cm from the AM;
- Patients underwent anterior resection of the rectum with curative intent, between March 2016 and July 2019;

- With preoperative MRI.

The exclusion criteria were as follows:

- Patients underwent more than one surgical procedure simultaneously;
- Urgent or emergent surgery;
- Presence of metastatic disease.

According to these criteria, a sample size of 73 patients was obtained (Figure 1).

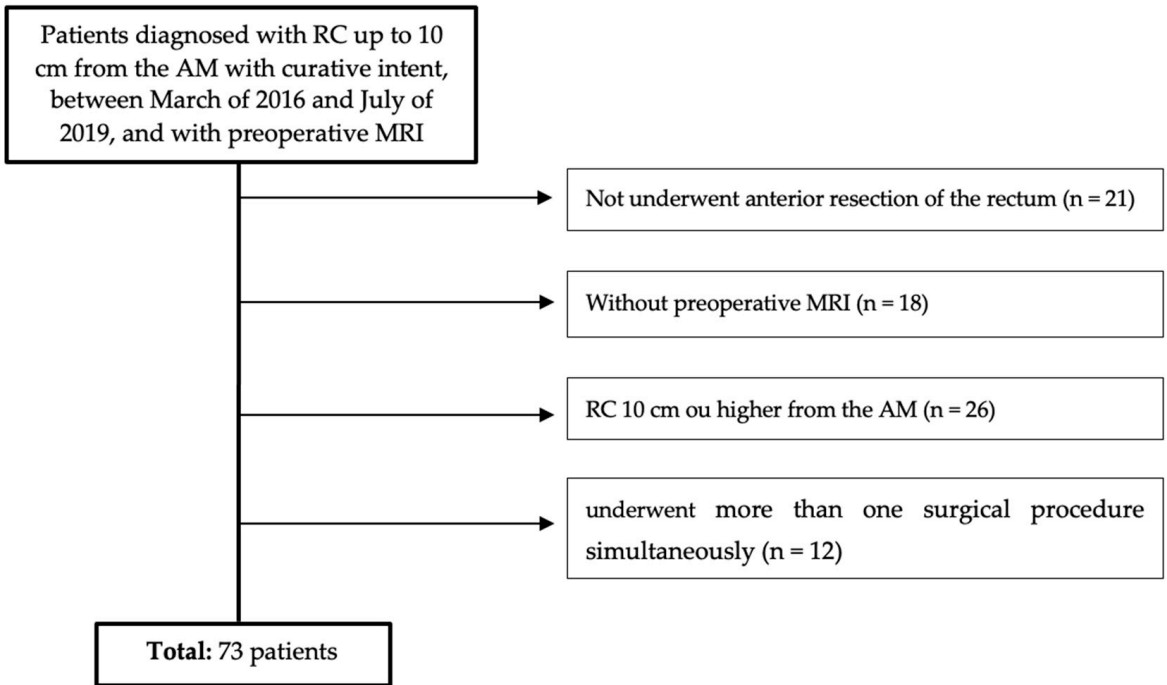

**Figure 1.** Flowchart of patient selection.

*2.1. Data Collection*

Data were collected through access to the informatics clinical processes in the Glintt Soluções Clínicas® and registered in a Microsoft Excel confidential document.

Variables related to patient demographic data (gender, age, BMI); pre-operatory data (preoperative therapy, level of tumor invasion, presence of positive nodes, degree of tumor differentiation and distance from the tumor to the AM in centimeters); operatory data (type of surgical approach and anastomosis performed, duration of surgery in minutes, conversion to laparotomy, transanal dissection, hemorrhage in milliliters); and postoperative data (postoperative hospital stay in days, postoperative morbidity (grade I, II, IIIa, IIIb, IVa, IVb and V), according to the classification proposed by Dindo et al. [14], and various pelvic measurements (transverse diameter, interspinous distance, ITD, distance from first sacred vertebra to the pubic symphysis, distance from the first sacred vertebra to the coccyx, distance from the pubic symphysis to the coccyx and MFA), all in cm.

The pelvic measurements on MRI (Figures 2–7) were carried out by the author, and to maintain the reproducibility of the results, measurements were taken, whenever possible, at the same anatomical levels.

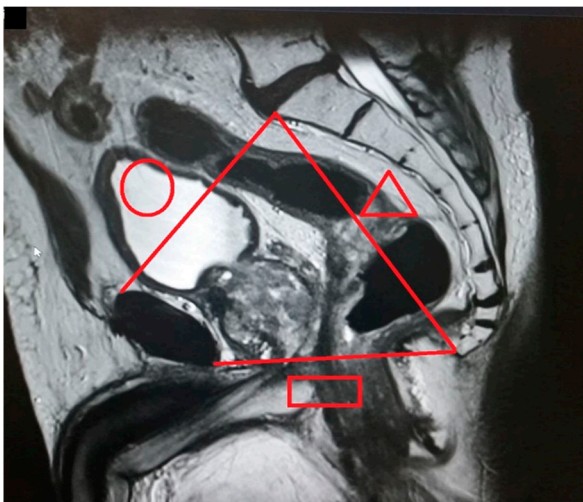

**Figure 2.** Pelvic inlet (circle): distance from the superior aspect of the pubic symphysis to the promontory). Pelvic outlet (rectangle): distance from the inferior aspect of the pubic symphysis to the tip of the coccyx. Pelvic depth (triangle): distance from the promontory to the tip of the coccyx.

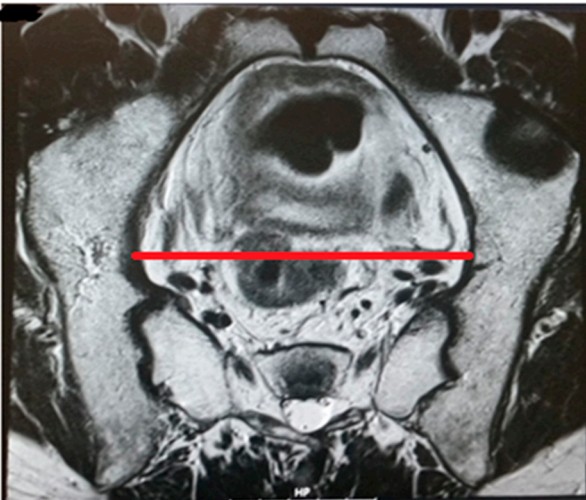

**Figure 3.** Transverse diameter.

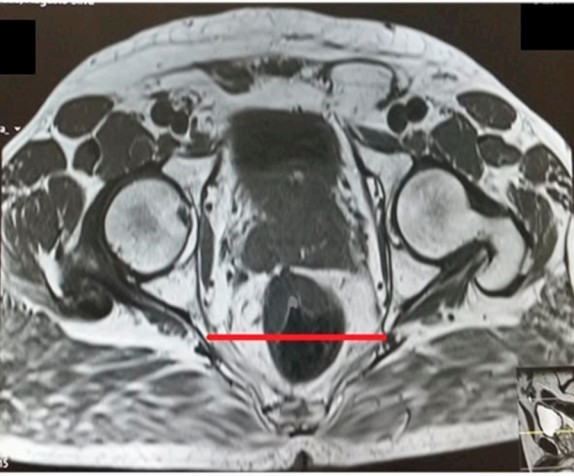

**Figure 4.** Interspinous distance.

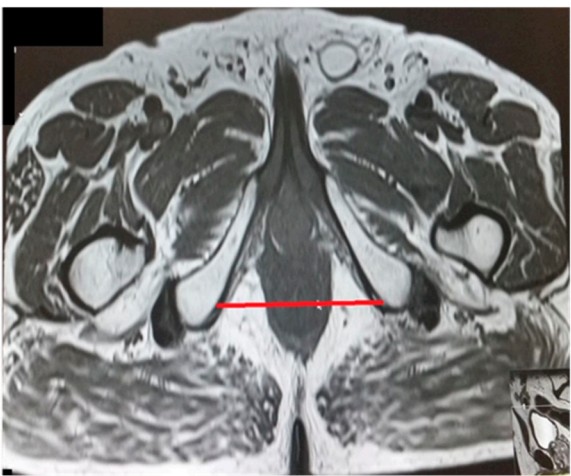

**Figure 5.** Intertuberous distance.

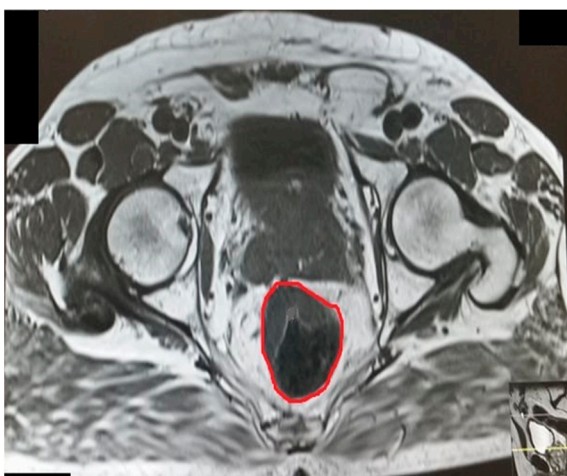

**Figure 6.** Rectal area.

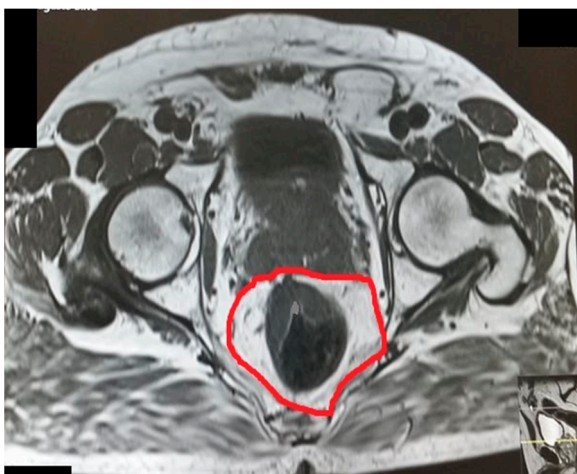

**Figure 7.** Mesorectal area.

## 2.2. Statistical Analysis

The statistical analysis was performed using the Statistical Package for Social Sciences (SPSS®) version 25.0.

A descriptive statistical analysis of the collected variables was carried out, and the relative and absolute frequencies of the categorical variables were described. The normality of the distribution of quantitative variables was assessed using the Shapiro–Wilk tests, skewness and kurtosis and by analyzing the histogram. Variables with normal distribution were described using the mean and standard deviation. Otherwise, the median (Mdn) and interquartile range (IQR) were used [15].

In order to validate the surgical difficulty prediction model proposed in the literature, each patient was assigned a surgical difficulty value according to the method described in the introduction. Then, the predictive model, $z = 0.99 \times \text{MFA} - 1.15 \times \text{CAA} - 1.05 \times \text{ITD} + 1.38 \times \text{BMI}$, was applied to each patient. As this model was based on logistic regression, by applying the equation $P = 1/(1 + e^{-z})$ it is possible to estimate the probability of the surgery being difficult or not. A probability greater than 50% was defined as a defining probability of an event (difficult surgery). In this way, a cross table was created with the predictive values and the real values, having estimated sensitivity and specificity for the predictive model.

To create a new predictive model, the surgical difficulty variable was dichotomized into low surgical difficulty or high surgical difficulty. Patients with a score of less than 6 on the surgical difficulty score were considered to belong to the first group and patients with a score of 6 or higher were considered to belong to the second group.

For the inferential analysis, the following variables were dichotomized: depth of tumor invasion was dichotomized into <T3 and ≥T3, and the variable number of lymph nodes involved was dichotomized into N0 and >N0.

Categorical variables were compared with the Chi-squared test ($x^2$). If the percentage of cells with an expected count below 5 was greater than 20%, Fisher's exact test was performed. For quantitative variables, the Student T test was used for independent samples. For the Student T tests, the assumptions of sample normality, homogeneity of variance and independence of the groups were verified. As an effect size measure for Fisher's exact test, the Phi coefficient (Φ) was used, and a small, medium and large effect was considered for values close to 0.10, 0.30 and 0.50, respectively. As a measure of effect size for the Student T tests, Cohen's d was reported, and a small, medium and large effect were considered for the approximate values of 0.20, 0.50 and 0.80, respectively [16].

The variables that presented statistically significant results, $p < 0.05$, were included in a binary logistic regression, with the aim of creating a predictive model of surgical difficulty [17]. Multicollinearity was assessed using a correlation matrix [17]. Using this model, the predicted probabilities were calculated and used to construct a Receiver Operating Characteristic (ROC) curve to determine the cutoff with maximum sensitivity, in order to equal the specificity value obtained in the literature. An Area Under the Curve (AUC) value close to 1 was considered a high probability of correct identification of individuals, while values of 0.5 were assumed to have no discriminatory capacity [17].

### 2.3. Ethical Considerations

This project required the collection of confidential data regarding the patients included in the sample. They were properly codified, and their anonymity is guaranteed.

This study received the approval of all the necessary institutions: the Ethics Committee for Health of Braga's Hospital (protocol code: 179_2019; date of approval: 11 September 2019), the Data Protection Office of Braga's Hospital (protocol code: 20190167_CirGeral290819; date of approval: 29 August 2019) and the Ethics Committee for Research in Life and Health Sciences (protocol code: CEICVS 124/2019; date of approval: 30 October 2019).

## 3. Results

A total of 74 patients were collected after applying the inclusion and exclusion criteria. The majority of the sample was male patients (68.5%), with a mean age of 64 years. The remaining demographic and preoperative data can be found in Table 2.

**Table 2.** Demographic and preoperative data.

|  | Total of Patients, n = 73 |
|---|---|
| Gender (Male/Female), n (%) | 50 (68.5%):23 (31.5%) |
| Age (years) | 64 (37–88) |
| Body Mass Index (kg/m$^2$) | 25.6 (17.3–37.3) |
| **Preoperative therapy**, n (%) |  |
| Radiochemotherapy | 47 (64.4%) |
| No preoperative therapy | 26 (35.6%) |
| **Tumor differentiation**, n (%) |  |
| Well differentiated | 37 (50.7%) |
| Moderately differentiated | 18 (24.7%) |
| Poorly differentiated | 1 (1.4%) |
| **Pathological T category**, n (%) |  |
| ypT0 | 18 (24.7%) |
| ypT1 | 13 (17.8%) |
| ypT2 | 22 (30.1%) |
| ypT3 | 18 (24.7%) |
| ypT4 | 2 (2.7%) |
| **Pathological N category**, n (%) |  |
| ypN0 | 57 (78.1%) |
| ypN1 | 12 (16.4%) |
| ypN2 | 4 (5.5%) |

After applying the normality tests, and evaluating the asymmetry, kurtosis and respective histogram, it was concluded that the pelvic measurements on the MRI followed a normal distribution. The values are described in Table 3.

**Table 3.** Pelvic measurements.

|  | Mean ± Standard Deviation |
|---|---|
| Interspinous distance (cm) | 9.70 ± 1.17 (7.45–12.4) |
| Intertuberous distance (cm) | 9.25 ± 1.24 (7.05–12.0) |
| Transverse diameter (cm) | 11.2 ± 1.33 (7.72–14.7) |
| Pelvic inlet (cm) | 10.7 ± 1.07 (7.9–13.0) |
| Pelvic depth (cm) | 12.3 ± 1.21 (10.1–14.6) |
| Pelvic outlet (cm) | 8.94 ± 1.01 (6.92–12.4) |
| Mesorectal fat area (cm$^2$) | 20.7 ± 6.36 (4.80–33.9) |

Surgical and postoperative data are reported in Table 4.

**Table 4.** Surgical and postoperative data.

|  | Total of Patients, n = 73 |
|---|---|
| **Surgical technique**, n (%) |  |
| Open | 19 (26.0%) |
| Laparoscopic | 54 (74.0%) |
| **Surgical procedure**, n (%) |  |
| Colorectal anastomosis | 46 (63.0%) |
| Coloanal anastomosis | 13 (17.8%) |
| Terminal colostomy | 14 (19.2%) |
| **Protective ileostomy**, n (%) |  |
| Sim | 22 (30.1%) |
| Não | 51 (69.9%) |

**Table 4.** *Cont.*

|  | Total of Patients, n = 73 |
|---|---|
| **Conversion to open procedure**, n (%) |  |
| Yes | 3 (4.1%) |
| No | 70 (95.9%) |
| **Transanal dissection,** n (%) |  |
| Yes | 5 (6.8%) |
| No | 68 (93.2%) |
| **Duration of surgery** (Minutes; M ± SD) | 264.4 ± 8.5 |
| **Blood loss**, n (%) |  |
| <200 mL | 56 (76.7%) |
| >200 mL | 17 (23.3%) |
| **Postoperative hospital stay** (Days; Mdn; IQR) | 6; 10 |
| **Morbidity grade**, n (%) |  |
| No morbidity | 37 (50.7%) |
| I | 7 (9.6%) |
| II | 9 (12.3)% |
| IIIa | 12 (16.4%) |
| IIIb | 3 (4.1%) |
| IVa | 3 (4.1%) |
| IVb | 1 (1.4%) |
| V | 1 (1.4%) |

IQR, interquartile range; Mdn, median; M, mean; SD, standard deviation.

### *3.1. Predictive Model Validation*

Of the 73 analyzed patients, 11 (15.1%) scored more than 5 in the surgical difficulty criteria and were therefore classified as high difficulty, with the remaining 62 (84.9%) as low difficulty.

After applying the equation $P = 1/(1 + e^{(-z)})$, it was possible to verify that the model predicted that 14 patients would have a high degree of surgical difficulty.

Next, it was necessary to check whether the predicted cases coincided with the real ones. Thus, the data, when compared with the real data in a double-entry table (Table 5), allowed us to infer that, in the population of patients with RC who underwent tumor resection surgery with the curative intent, the sensitivity and specificity of the model are 18% and 81%, respectively.

**Table 5.** Crossing between the predictive model and the real difficulty of the sample.

|  | Predictive Score < 6 | Predictive Score ≥ 6 |
|---|---|---|
| Real score < 6 | 50 | 12 |
| Real score ≥ 6 | 9 | 2 |

### *3.2. Creating a New Predictive Model*

As can be seen, given the sensitivity and specificity values of the existing predictive model, it is not possible to reliably assess which patients will have high surgical difficulties. Given this, it became imperative to develop a new proposal for a predictive model of surgical difficulty, based on the variables described previously. Thus, low surgical difficulty was considered for scores lower than 6 and high surgical difficulty for scores of 6 or higher [11].

To verify the existence of associations between the independent variables and surgical difficulty, univariate analysis was performed.

The results of the univariate analysis between surgical difficulty and demographic and preoperative data are shown in Table 6. As can be seen, there were statistically significant

differences in relation to gender (*Phi* = −0.286; *p* = 0.013). When we analyzed adjusted residuals, male patients were the group with the most difficult surgeries.

**Table 6.** Univariable logistic regression analysis of associations between demographic and preoperative data and risk of surgical difficulty.

|  | Low-Risk Group (n = 62) | High-Risk Group (n = 11) | *p*-Value |
|---|---|---|---|
| **Gender** |  |  |  |
| Male | 39 (53.4%) | 11 (15.1%) | *p* = 0.013 |
| Female | 23 (31.5%) | 0 (0.0%) |  |
| **Age** (Years; M ± SD) | 65.3 ± 12.3 | 58.6 ± 11.9 | *p* = 0.095 |
| **BMI** (kg/m$^2$; M ± SD) | 25.5 ± 3.59 | 26.5 ± 3.73 | *p* = 0.403 |
| **Preoperative therapy** |  |  |  |
| Radiochemotherapy | 40 (54.8%) | 7 (9.6%) | *p* = 1.000 |
| No preoperative therapy | 22 (30.1%) | 4 (5.5%) |  |
| **Distance to the anal margin** (cm; M ± SD) | 7.13 ± 2.50 | 7.86 ± 1.90 | *p* = 0.365 |
| **Pathological T category** |  |  |  |
| <ypT3 | 44 (60.3%) | 9 (12.3%) | *p* = 0.716 |
| ≥ypT3 | 18 (24.7%) | 2 (2.7%) |  |
| **Lymph Node—Positive** |  |  |  |
| ypN0 | 48 (65.8%) | 9 (12.3%) | *p* = 1.000 |
| >ypN0 | 14 (19.2%) | 2 (2.7%) |  |

BMI, body mass index; M, mean; SD, standard deviation.

Student T tests for independent samples revealed the existence of strong associations and statistically significant associations between the outcome and the variables transverse diameter of the pelvis (*t* = 3.610, *p* < 0.001, d = 1.02), ISD (*t* = 2.522, *p* = 0.014, d = 0.92) and ITD (*t* = 4.121, *p* < 0.001, d = 1.01). No other results were statistically significant, as described in Table 7.

**Table 7.** Univariable logistic regression analysis of associations between pelvimetry and risk of surgical difficulty.

|  | Low-Risk Group (n = 62) | High-Risk Group (n = 11) | *p*-Value |
|---|---|---|---|
| **Interspinous distance** (cm; M ± SD) | 9.84 ± 1.17 | 8.91 ± 0.81 | *p* = 0.014 |
| **Intertuberous distance** (cm; M ± SD) | 9.40 ± 1.26 | 8.40 ± 0.60 | *p* < 0.001 |
| **Transverse diameter** (cm; M ± SD) | 11.4 ± 1.15 | 9.97 ± 1.62 | *p* < 0.001 |
| **Pelvic inlet** (cm; M ± SD) | 10.7 ± 1.12 | 10.7 ± 0.74 | *p* = 0.774 |
| **Pelvic depth** (cm; M ± SD) | 12.2 ± 1.20 | 12.8 ± 1.19 | *p* = 0.109 |
| **Pelvic outlet** (cm; média ± M ± SD) | 9.00 ± 0.96 | 8.62 ± 0.96 | *p* = 0.257 |
| **Mesorectal fat area** (cm$^2$; M ± SD) | 20.9 ± 6.30 | 19.6 ± 6.86 | *p* = 0.517 |

M, mean; SD, standard deviation.

The surgical data introduced in the univariate analysis were the type of surgical approach, type of anastomosis performed and protective ileostomy. The results can be seen in Table 8. There were no statistically significant differences in the analysis of surgical variables.

**Table 8.** Univariable logistic regression analysis of associations between surgical data and risk of surgical difficulty.

| | Low-Risk Group (n = 62) | High-Risk Group (n = 11) | *p*-Value |
|---|---|---|---|
| **Surgical technique** | | | |
| Open | 18 (24.7%) | 1 (1.4%) | *p* = 0.268 |
| Laparoscopic | 44 (60.3%) | 10 (13.7%) | |
| **Surgical procedure** (n = 59; LD = 48; HD = 11) | | | |
| Colorectal anastomosis | 38 (64.4%) | 8 (13.6%) | *p* = 0.693 |
| Coloanal anastomosis | 10 (16.9%) | 3 (5.1%) | |
| **Protective ileostomy** | | | |
| Yes | 17 (23.3%) | 5 (6.8%) | *p* = 0.289 |
| No | 45 (61.6%) | 6 (8.2%) | |

HD, high difficulty; LD, low difficulty.

The variables statistically significant in the univariate analysis, gender ($p = 0.013$), transversal diameter ($p < 0.001$), ITD ($p < 0.001$) and ISD ($p = 0.014$), were included in the binary logistic regression. The variable gender had a very high standard error, which can be easily justified given that all patients with high surgical difficulty were male. Therefore, the logistic regression was performed without including the gender variable.

The predictive model allowed us to estimate the surgical difficulty using the equation $z = 11.2 - 0.001 \times \text{ISD} - 0.045 \times \text{ITD} - 0.084 \times \text{transversal diameter}$ (Table 9). This model was statistically significant with a *p* value of 0.004.

**Table 9.** Multivariable logistic regression analysis.

| Variables | B | S.E | Wald | *p* | O.R | 95% CI |
|---|---|---|---|---|---|---|
| Constant | 11.270 | 4.491 | 6.297 | 0.012 | 78417.9 | |
| Interspinous distance | −0.001 | 0.052 | 0.000 | 0.991 | 0.999 | 0.903–1.107 |
| Intertuberous distance | −0.045 | 0.051 | 0.777 | 0.378 | 0.956 | 0.866–1.056 |
| Transversal diameter | −0.084 | 0.042 | 3.944 | **0.047** | 0.919 | 0.846–0.999 |

B, Coefficient; CI, Confidence interval; O.R, Odds ratio; S.E, Standard error.

The transverse diameter was the pelvic measurement that most contributed to the prediction of high surgical difficulty, due to the highest Wald value (3.944). When the transverse diameter decreases, surgical difficulty increases (O.R. 0.919, 95% I.C. 0.846–0.999, $p = 0.047$). The remaining variables were not statistically significant.

We can confirm that, based on data from our patients, the predictive model created has a sensitivity of 18% and a specificity of 97%, while the predictive model already existing in the literature has a sensitivity of 18% and a specificity of 81% (Table 10).

**Table 10.** Sensitivity and specificity of our predictive model.

| | Predictive Score < 6 | Predictive Score ≥ 6 |
|---|---|---|
| Real score < 6 | 60 | 2 |
| Real score ≥ 6 | 9 | 2 |

In order to improve the sensitivity of the model, a ROC curve was created (Figure 8).

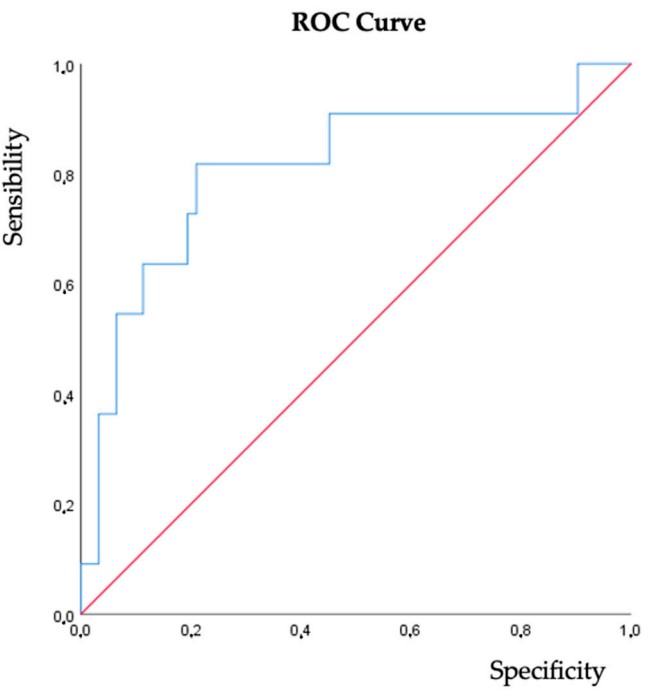

**Figure 8.** ROC curve—predictive model of surgical difficulty.

The AUC of the ROC curve is 0.809 (95% CI 0.649–0.969), *p* = 0.001, and the proposed cutoff is 0.198, for which the sensitivity is 73% and the specificity is 81%.

## 4. Discussion

Low-quality tumor excision is the main cause of local recurrence of CR, which may be due to surgeries with a high degree of difficulty [18]. Surgeries in this anatomical region can often be complicated, mainly due to the small size of the pelvis [10,13]. Thus, this retrospective study with a sample of 73 patients, all of them with tumors with a maximum distance from the AM of 10 cm, attempts to assess the influence of pelvic measures in surgical difficulty on the patients submitted to tumoral surgical resection with curative intent and whether the current predictive models are effective.

About 15% of the patients had highly difficult surgeries, and all of them were male. In the univariate analysis, statistically significant differences were observed regarding gender (*p* = 0.013). Men have smaller pelvic dimensions compared to women, which may eventually justify the statistically significant differences between gender and surgical difficulty, suggesting that small pelvis dimensions are associated with more complicated surgeries [18]. However, not all pelvic measurements contribute equally to surgical difficulty [13]. A univariate analysis of pelvic measurements collected from preoperative MRI was then performed. The results showed the existence of statistically significant differences between surgical difficulty and the transverse diameter of the pelvis (*p* < 0.001), the IED (*p* = 0.014) and the ITD (*p* < 0.001). Previous studies regarding the effect of pelvimetry in predicting surgical difficulty also indicate these same measures as those with the greatest impact on surgical difficulty [7,8].

BMI has been widely described in the literature as an aggravating risk factor for surgical difficulty [19,20], but our study did not show a statistically significant difference. This outcome may be due to the great heterogeneity of the sample included and given the fact that only seven patients had a BMI greater than 30 Kg/m$^2$. Similarly in our study, using neoadjuvant radiochemotherapy, a tumor closer to the AM and tumor staging did not significantly affect surgical difficulty [9,10,21,22].

Yamamoto et al. indicated that high-grade surgical difficulty was associated with a pelvic outlet of less than 82.7 mm [23]. In our multivariate analysis, only the transverse diameter of the pelvis was identified as an independent risk factor (O.R. 0.919, 95% I.C.

0.846–0.999, *p* = 0.047). Thus, a surgical difficulty model was created, z = 11.2 − 0.001 × ISD − 0.045 × ITD − 0.084 × transverse diameter, with a sensitivity of 18% and a specificity of 97%.

This predictive model faced obstacles to its application. Although this model seems better than the one already available in the literature, with better specificity of 97% vs. 81% [11], it means that it only identifies a higher percentage of cases with less surgical difficulty. Furthermore, the fact that it was created based on an outcome that included only male patients makes it essential to exercise caution when extrapolating the results to the female population.

Therefore, a ROC curve was created in order to improve the sensitivity of this model. A cutoff point was established at 0.198, with a value greater than 0.198 identifying 72.7% of patients who will undergo high-difficulty surgery and, on the contrary, a value lower than 0.198 correctly identifies 80.6% of patients who will undergo low-difficulty surgery. In this way, it was possible to present a substantially better model than the one already existing in the literature.

One of the limitations of this study is its retrospective nature and the subjectivity of evaluating surgical difficulty, so the occurrence of information bias is possible. Therefore, future studies, preferably prospective, are necessary in order to associate the surgeon's opinion and his expertise with the assessment of surgical difficulty. Another limitation is the single-center design, with a sample size that was relatively small and heterogeneous.

In conclusion, if pelvimetry proves to be a predictor of surgical difficulty, it would be interesting to implement protocols that provide, in a systematic way, the measurement of transverse diameter, ISD and ITD from pre-operatory MRI in patients with RC. This would allow us to identify the patients with high surgical difficulty, providing more specialized and individualized monitoring for these patients.

**Author Contributions:** Conceptualization, J.S., P.M.D.d.S., C.C.P. and S.F.M.; Methodology, P.M.D.d.S., C.C.P. and S.F.M.; Validation, P.M.D.d.S., C.C.P. and S.F.M.; Formal Analysis, P.M.D.d.S. and J.S.; Investigation, P.M.D.d.S. and J.S.; Resources, P.M.D.d.S. and J.S.; Writing—Original Draft Preparation, J.S and P.M.D.d.S.; Writing—Review and Editing, P.M.D.d.S., C.C.P. and S.F.M.; Supervision, P.M.D.d.S., C.C.P. and S.F.M. All authors have read and agreed to the published version of the manuscript.

**Funding:** This research received no external funding.

**Institutional Review Board Statement:** This study was conducted in accordance with the Declaration of Helsinki and approved by all the necessary institutions: the Ethics Committee for Health of Braga's Hospital (protocol code: 179_2019; date of approval: 11 September 2019), the Data Protection Office of Braga's Hospital (protocol code: 20190167_CirGeral290819; date of approval: 29 August 2019), and the Ethics Committee for Research in Life and Health Sciences (protocol code: CEICVS 124/2019; date of approval: 30 October 2019).

**Informed Consent Statement:** Informed consent was obtained from all subjects involved in the study.

**Data Availability Statement:** The data presented in this study are available on request from the corresponding author. The data are not publicly available due to the protection of patient privacy.

**Conflicts of Interest:** The authors declare no conflicts of interest.

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
