# Peer review of "Assessment of Surgical Difficulty in Patients with Rectal Cancer—The Impact of Pelvimetry"

_2673-8937, doi:10.3390/ijtm4010009_

Round 1
Reviewer 1 Report
Comments and Suggestions for Authors
Thank you for permitting me to review this manuscript
In this work the authors assessed the influence of pelvic measures obtained by MRI in surgical difficulty
using adequate modelisation the authors declare a sensitivity for the transverse diameter of the pelvis of 18% and a specificity of 97% in comparison to the litterature with the same sensitivity but a specificity of 81% in male patients only
Abstrtact conclusion I don't think guaranty is the appropriate term in this context I would rather say predict
The authors need to include more recent articles in the subject for ex prediction of surgical difficulty in minimally invasive surgery for rectal cancer by use of MRI pelvimetry
T. Yamamoto, et al...August 2020, Pages 666–677, https://doi.org/10.1002/bjs5.50292
and compare their results
Results : since 18 patients did not have MRI how this was manged in the results ?
Line 335 to 339 please describe better the obstacles to application ,, why predicting less surgical difficulty is an obstacle ?
Line 347 while the authors introduced a table with objective according to litterature different scale of surgical difficulty , why they used a subjective way to assess surgical difficulty (line 347?)
Comments on the Quality of English Language
the study need minor english adjustements for ex
measurements of pelvic measurements ....
Author Response
First of all, thank you for your comments! I have already made the changes you suggested and compared our results with recent bibliography.
"since 18 patients did not have MRI how this was manged in the results ?" Answer: These patients were excluded (see Material and Methods)
"Line 335 to 339, further describe the obstacles to application, why is predicting less surgical difficulty an obstacle?" Answer: Our main objective is to identify the majority of cases with high surgical difficulty. Therefore, a ROC curve was created in order to improve the sensitivity of our model.
Thank you!
Reviewer 2 Report
Comments and Suggestions for Authors
The authors investigate of Surgical Difficulty in Patients with Rectal Cancer in terms of the Impact of Pelvimetry. The study is so interesting, however, I have some concerns to discuss.
Whats is the novelty of the current study?
Please compare cases with small and large pelvic diameters.
Even in cases without small pelvic diameters, careful planning may be necessary.
What specific plans are needed?
Author Response
First of all, thank you for your comments!
The main aim was to study the influence of pelvic measures in surgical difficulty on the patients submitted to tumoral surgical resection with curative intent and to what extent this preoperative data can be useful for surgeons.
This will allow identifying patients with high surgical difficulty, providing a more intensive and rigorous follow-up for these patients in the postoperative period (e.g. identification of anastomotic dehiscence, ...). It may also be useful to decide what type of approach (open Vs laparoscopic Vs robotic) will be performed
Thank you
Reviewer 3 Report
Comments and Suggestions for Authors
The authors in this study retrospectively study examined the influence of pelvic measures in surgical difficulty on the patients submitted to tumoral surgical resection with curative intent.
The authors should be commended on this single institution effort.
The study is methodologically well performed, but the sample size is relatively small and heterogeneous.
Whether all patients were operated on by the same surgeon?
How do you explain that preoperative radio chemotherapy was carried out in 64.4% of patients and that after surgery in 21.9% of patients it was present N1, N2?
The findings of this study are significant for clinical practice.
Author Response
Thank you for your comments!
The patients were operated only by the Unit of Coloproctology.
The findings of this study can change the clinical practice and improve the surgical and oncological outcomes
Thank you!
Round 2
Reviewer 2 Report
Comments and Suggestions for Authors
The authors have replied well, so the manuscript is suitable for publication.